# Improving Hyperspectral Adversarial Robustness Under Multiple Attacks

**Nicholas Soucy & Salimeh Yasaei Sekeh**
Department of Computer Science, SCIS, University of Maine
{nicholas.soucy,salimeh.yasaei}@maine.edu

## Abstract

Semantic segmentation models classifying hyperspectral images (HSI) are vulnerable to adversarial examples. Traditional approaches to adversarial robustness focus on training or retraining a single network on attacked data, however, in the presence of multiple attacks these approaches decrease in performance compared to networks trained individually on each attack. To combat this issue we propose an *Adversarial Discriminator Ensemble Network* (ADE-Net) which focuses on attack type detection and adversarial robustness under a unified model to preserve per data-type weight optimally while robustifiying the overall network. In the proposed method, a discriminator network is used to separate data by attack type into their specific attack-expert ensemble network.

## 1 Introduction

Current semantic segmentation models are vulnerable to the addition of imperceptible perturbations to the input data Biggio et al. (2013). These perturbations are well-crafted attacks that when added in small amounts to a sample, drastically fool the model and decrease accuracy Szegedy et al. (2013b). To combat this challenge and make the model robust toward attacks, adversarial training is performed on the model. To perform adversarial training on a model, one generates the adversarial examples and then either continually trains the trained model, or mixes the attack examples with non-attacked examples and trains a new model from scratch Goodfellow et al. (2014b); Szegedy et al. (2013c). In this approach, the model performs worse on each individual data attack type due to the updated weights, but better overall. Another approach is to detect the attacked data to avoid it during classification via adversarial detection. In this case, the attacks are often ignored, despite having information that might be valuable to classify Metzen et al. (2017). In this work, we fuse the strengths of these two approaches. To detect and defend against adversarial attacks in one model, a novel Adversarial Discriminator Ensemble Network (ADE-Net) model is proposed. While most models focus on network architecture and robustness, ADE-Net can use any semantic segmentation network while classifying attack type and class type in one unified model.

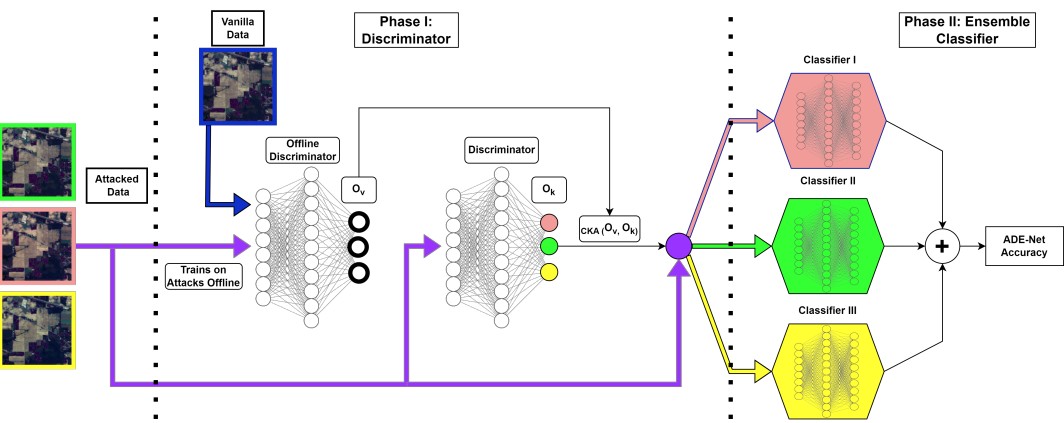

Figure 1: Overview of ADE-Net model structure: **Phase I** contains the Discriminator Network and **Phase II** contains the Ensemble Networks.

## 2 PROPOSED METHOD: ADE-NET

**Notation:** Consider vanilla data as $\{x_i^v, y_i^v\}_{i=1}^{n_v}$ with size $n_v$. Define attack label $k = 1, \ldots, K$ and consider $k$-th attack data $\{x_i^k, y_i^k\}_{i=1}^{n_k}$ with size $n_k$. Each attacked point has two labels: class label and attack label, $\{x_i^k, y_i^k, c_i = k\}_{i=1}^{n_k}$. For a classifier $F$, consider a discriminator $F_D$ that focuses on separating input based on attack type, and ensemble classifiers $F_k$, which denotes the $k$-the classifier expert in attack type $k$. For each ensemble network, $\omega_k^*$ is the optimal weight of $F_k$, and $\omega_D^*$ is the optimal weight of $F_D$ on all $K$ attack data. Let $O_v \in \mathcal{R}^{m \times n_v}$ be the logits of passing vanilla on an offline trained $F_D^{offline}$ on attacks, and $O_k \in \mathcal{R}^{m \times n_k}$ be the logits of $F_D$ on single attack $k$.

**Methodology:** For ADE-Net, our overall model structure consists of two phases: 1) Discriminator Phase I: Adds adversarial detection to ADE-Net via attack type classification and CKA Kornblith et al. (2019) to separate mixed attack data. 2) Ensemble Phase II: Constituted by a number of networks, equal to $n_k$, that focus on regular class label classification. Each network is an expert at a particular attack determined by the output label from the discriminator. Figure 1 presents a visual overview of ADE-Net. We propose in ADE-Net that the ensemble classifiers are trained collaboratively with the discriminator focusing on class and attack classification respectively:

$$\omega_D^*, \omega_k^* = \arg \min_{\omega_D, \omega_k} \sum_{k=1}^{K} \alpha_k \mathcal{L}_{\omega_k}(F_k(\mathbf{x}), y) + \sum_{k=1}^{K} \lambda_k \, CKA(O_v, O_k) + \beta \, \mathcal{L}_{\omega_D}(F_D(\mathbf{x}), c) \quad (1)$$

where the first $\alpha_k$ is for Phase II and the remaining $\lambda_k$ and $\beta$ terms are for Phase I. Extended mathematical formalization of the loss function can be found in the Appendix in Section C. The overview of our ADE-Net algorithm is shown in Algorithm 1.

| # Attack | All = 1.0 | $\lambda_k=\beta=0.1$ $\alpha_k=10.0$ | $\lambda_k=\beta=10.0$ $\alpha_k=0.1$ | $\lambda_k=\beta=1.0$ $\alpha_k$ Attack Dependent | $\lambda_k=0$ (No CKA) $\beta=\alpha_k=1.0$ | Baseline |
|---|---|---|---|---|---|---|
| 2 | I: **95.29 ± 0.01** 
 II: **80.37 ± 0.01** | I: 95.19 ± 0.04 
 II: 80.00 ± 0.01 | I: 95.34 ± 0.03 
 II: 80.11 ± 0.02 | I: 95.21 ± 0.04 
 II: 80.30 ± 0.01 | I: 95.00 ± 0.05 
 II: 79.67 ± 0.02 | **78.78 ± 0.08** |
| 3 | I: 63.97 ± 0.03 
 II: 81.57 ± 0.02 | I: 64.12 ± 0.04 
 II: 81.71 ± 0.02 | I: 63.97 ± 0.04 
 II: 81.62 ± 0.03 | I: **63.99 ± 0.03** 
 II: **81.72 ± 0.02** | I: 63.50 ± 0.04 
 II: 81.49 ± 0.04 | **80.11 ± 0.13** |
| 4 | I: 48.03 ± 0.04 
 II: 82.23 ± 0.12 | I: 47.98 ± 0.10 
 II: 82.06 ± 0.11 | I: 47.98 ± 0.09 
 II: 82.1 ± 0.12 | I: **48.05 ± 0.05** 
 II: **82.51 ± 0.09** | I: 47.51 ± 0.10 
 II: 81.66 ± 0.14 | **81.37 ± 0.02** |
| 5 | I: 50.30 ± 0.03 
 II: 83.27 ± 0.15 | I: 50.40 ± 0.02 
 II: 83.27 ± 0.14 | I: 50.45 ± 0.03 
 II: 83.17 ± 0.16 | I: **50.30 ± 0.03** 
 II: **83.55 ± 0.13** | I: 50.40 ± 0.04 
 II: 83.10 ± 0.16 | **82.65 ± 0.22** |

Table 1: Experimental results of ADE-Net with Indian Pines dataset. Best performing results for each row are in bold, worst-performing is in red. The $\alpha_k$ hyperparameter is for the ensemble class cross-entropy, $\lambda_k$ is for the discriminator CKA and $\beta$ is for the discriminator attack cross-entropy.

## 3 EXPERIMENTS

For our experiments, we use the HSI dataset Indian Pines (IP). Two additional datasets, Kennedy Space Center (KSC) Amigo (2020), and Houston Labate et al. (2019) are available in Appendix Section D. Two averaged overall accuracies (OA) terms will be reported: Phase I and Phase II accuracy as shown in Figure 1. Phase I Discriminator accuracy captures attack-label classification while Phase II Ensemble accuracy captures class-label accuracy.

**Evaluation:** We showcase different hyperparameter values to find the experimentally optimal values for better ADE-Net accuracy. We also explore the role of CKA by removing it in certain experiments. Results are found in Table 1 for IP. For all numbers of attacks, ADE-Net outperforms the baseline. The discriminator performs worse when more attacks are added, however, with the addition of vanilla data in our five attack experiments, we can see the discriminator performing better due to the difference between attacked and non-attacked data. The highest Phase II accuracy is achieved in our attack dependent $\alpha_k$ tests, showing that giving more attention to more difficult attacks increases the overall accuracy of ADE-Net. The No CKA ($\lambda_k = 0$) tests perform worse than all the others, showing the positive effect CKA has on ADE-Net.

**Conclusion:** We analyzed the effect that multiple adversarial attacks have on HSI semantic segmentation. An approach was developed leveraging attack-type detection and robustness in one unified network: ADE-Net. Though a Phase I discriminator attack-type classifier leveraging the similarity measure CKA, and a Phase II attack-expert ensemble network, ADE-Net outperformed the single model baseline for all datasets with all attack combinations.

URM STATEMENT

The authors acknowledge that at least one key author of this work meets the URM criteria of ICLR 2023 Tiny Papers Track.

ACKNOWLEDGEMENTS

This work has been partially supported NSF 1920908 (EPSCoR RII Track-2) and CAREER-NSF 5409260; the findings are those of the authors only and do not represent any position of these funding bodies.

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

## A  RELATED WORKS

In Szegedy et al. (2013a) the authors showed that adding a small well-crafted perturbation to a sample can be catastrophic to a network's performance. These adversarially attacked images often look the same to a human observer but make the network vulnerable. The study of adversarial examples and robustness in HSI semantic segmentation is a new entry into the field with Xu et al. (2021a) being the first to introduce the idea of adversarial attack-defense in the HSI domain. They created a self-attention context network (SACNet) to better defend against these attacks. The authors in Xu et al. (2021b) use their Masked Spatial-Spectral Autoencoder (MSSA) which consists of masked sequence attention learning, dynamic graph embedding, and self-supervised reconstruction. Rather than focusing on semantic segmentation via network architecture, some works like Park et al. (2021); Qi et al. (2022) try using rich spectral information to robustify the entire process. In Park et al. (2021), they propose a spectral sampling and shape encoding to increase adversarial robustness as a preprocessing step to traditional per-pixel classification via random sampling. However, all these approaches focus on a single attack at a time and do not explore network robustness in the presence of multiple attacks. Recently, AutoAttack Croce & Hein (2020) extends PGD to create an aggregate attack and achieves lower robust networks. A union of attacks similar to AutoAttack is created in Maini et al. (2020) by using a generalized PGD-based procedure to incorporate multiple perturbation models into a single attack which also leads to drastically worse performance than individual perturbations. Many existing works use these aggregate attacks now to test performance, but still, test one aggregate attack at a time.

## B  CENTERED KERNEL ALIGNMENT (CKA)

In ADE-Net, we use a network-layer similarity measure called Centered Kernel Alignment (CKA) Kornblith et al. (2019) as follows:

$$CKA(O_v, O_k) = \frac{\left\| (O_v)^T O_k \right\|_F^2}{\left\| (O_v)^T O_v \right\|_F^2 \left\| (O_k)^T O_k \right\|_F^2}, \quad (2)$$

For this particular use of CKA, we are using vanilla logits $O_v$ from a discriminator trained offline on attacks to determine the dissimilarity between different attacks. $O_k$ is calculated during training where $k$ is the assigned attack type for each sample during training by the discriminator. CKA

was chosen for its intuition over other similarity measures to aid our attack separation. Adversarial attacks have a prevalent effect on a network in deeper layers, therefore, once CKA calculates the relative dissimilarity between the logits of vanilla data and attacked data, we have an intuitive measure to use attack information in the logit space rather than just the feature space.

## C   ADE-NET EXTENDED FORMALIZATION

**Discriminator Phase I:** The aim of the discriminator is to add adversarial detection to ADE-Net via attack type classification and CKA to separate mixed attack data. For the discriminator, we use a categorical cross-entropy loss function as $\omega_D^* = \arg\min_{\omega_D} \mathcal{L}_{\omega_D}(F_D(\mathbf{x}))$. To increase attack classification accuracy further, we use the similarity measure CKA from 2 to incorporate more information about attacks into the loss function. This term is the sum of CKAs between logit-layer vanilla data output on a trained discriminator, and the logit-layer output of each attack present during training. The optimal $\omega_D^*$ is learned by solving the optimization problem: $\arg\min_{\omega_D} \beta \; \mathcal{L}_{\omega_D}(F_D(\mathbf{x})) + \sum_{k=1}^{K} \lambda_k \; CKA(O_v, O_k)$, where $\lambda_k$ and $\beta$ are hyperparameters and $\mathcal{L}_{\omega_D}$ is the cross-entropy function on attack labels while also incorporating the similarity of the attack compared to non-attack data in deeper layers of a trained network.

**Ensemble Phase II:** The ensemble phase is constituted by a number of networks, equal to $n_k$, that focus on regular class label classification. Each network is an expert at a particular attack determined by the output label from the discriminator. We use an ensemble categorical cross-entropy loss function $\omega_k^* = \arg\min_{\omega_k} \sum_{k=1}^{K} \alpha_k \mathcal{L}_{\omega_k}(F_k(\mathbf{x}), y)$, where $\alpha_k$, $k = 1, \ldots, K$ are hyperparameters and allow for individual networks to get more attention than others.

---

**Algorithm 1:** ADE-Net model

---

**Input:** Data $\{x_i, y_i\}_{i=1}^{N}$. Set $K$, $E$: # of attacks and epochs. Learning rates $\eta_D$ and $\eta_j$, $j = 1, \ldots, K$.

**Output:** Overall Test Accuracy

Attack data $\{x_i, y_i\}_{i=1}^{N}$ with $k$ attack types to get $\{x_i^k, y_i^k, c_i = k\}_{i=1}^{n_k}$ and vanilla data $\{x_i^v, y_i^v\}_{i=1}^{N}$.

Train Discriminator $F_D$ on attacked training data offline and pass vanilla and logits $O_v$. Shuffle attack data $\{x_i^k, y_i^k, c_i = k\}_{i=1}^{n_k}$ (including vanilla data).

**for** $e = 1, \ldots, E$ **do**

    **for** $b = batch_1, \ldots, batch_B$ **do**

        Input data into $F_D$ on the attack type and compute $CKA(O_v, O_k)$. Update $F_D$'s weights as $\omega_D \leftarrow \omega_D - \eta_D \nabla_{\omega_D}(\beta \; \mathcal{L}_{\omega_D}(F_D(\mathbf{x})) + \sum_{k=1}^{K} \lambda_k \; CKA(O_v, O_k))$

        Discriminate attacks based on the $\omega_D$.

        **for** $j = 1, \ldots, k$ **do**

            Input $\{x_i^j, y_i^j, c_i = j\}_{i=1}^{n_j}$ discriminated by $F_D$ into $F_j$. Update classifier $F_j$ weights as $\omega_j \leftarrow \omega_j - \eta_j \nabla_{\omega_j} \sum_{k=1}^{K} \alpha_k \mathcal{L}_{\omega_k}(F_k(\mathbf{x}), y)$

        **end**

    **end**

**end**

Report $Acc$

---

## D   EXPERIMENTAL PARAMETERS & EXTENDED EXPERIMENTS

In our main paper, we used the Indian Pines dataset. For extended experiments, we include the Kennedy Space Center (KSC) and Houston datasets. Experimental results for these extra datasets can be found in Table 2 for KSC and Table 3 for Houston. To better visualize the impact of CKA on ADE-Net, Figure 2 is shown to denote how CKA can effectively separate attacks in the logit layer during training.

For all of our experiments, each dataset is reduced to 30 bands using Principal Component Analysis (PCA). We use U-Net Ronneberger et al. (2015) for the discriminator and all ensemble networks. We use Adam optimizer with a learning rate of 0.001 for all networks under three trials. We use a batch size of 256 and train ADE-Net for 100 epochs. For attacks, we use the Fast Gradient Signed Method (FGSM) Goodfellow et al. (2014a), Carlini and Wagner (CW) Carlini & Wagner (2017), Projected Gradient Decent (PGD) Madry et al. (2017), and Iterative Fast Gradient Signed Method (I-FGSM) Xie et al. (2019). In our experiments each attack is added on at a time starting at, FGSM, then CW, PGD, I-FGSM, and finally non-attacked vanilla data. All attacks were generated using an $\epsilon = 0.1$. In the experiment where $\alpha_k = X_k$, as shown in Tables 1, 2 and 3, values are weighted by the relative difficulty for classification on that attack as: $\alpha_{FGSM} = 1.4$, $\alpha_{CW} = 2.3$, $\alpha_{PGD} = 1.7$, $\alpha_{I-FGSM} = 1.3$, and for vanilla $\alpha_{Vanilla} = 1.0$.

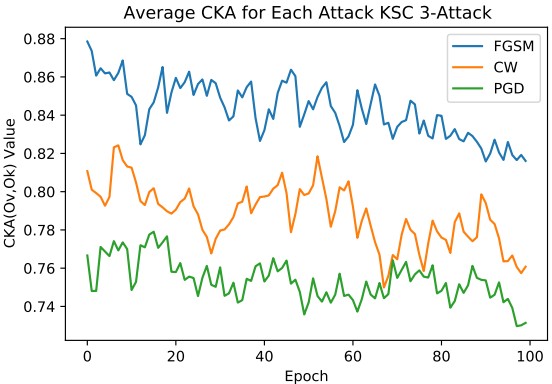

Figure 2: Comparison of attacks' CKA values calculated from the average of each batch for each FGSM, CW, and PGD attack during ADE-Net training for the KSC dataset. Note that for each epoch the three attacks are consistently separated.

| # Attack | All = 1.0 | $\lambda_k=\beta=0.1$ $\alpha_k=10.0$ | $\lambda_k=\beta=10.0$ $\alpha_k=0.1$ | $\lambda_k=\beta=1.0$ $\alpha_k$ Attack Dependent | $\lambda_k=0$ (No CKA) $\beta=\alpha_k=1.0$ | Baseline |
|---|---|---|---|---|---|---|
| 2 | I: $100.0 \pm 0.0$ 
 II $87.45 \pm 0.39$ | I: $100.0 \pm 0.0$ 
 II: $86.87 \pm 0.38$ | I: $100.0 \pm 0.0$ 
 II: $86.94 \pm 0.29$ | **I: $99.98 \pm 0.01$** 
 **II: $87.63 \pm 0.24$** | I: $99.54 \pm 0.02$ 
 II: $86.01 \pm 0.36$ | $83.59 \pm 0.40$ |
| 3 | **I: $98.3 \pm 0.04$** 
 **II: $88.07 \pm 0.04$** | I: $97.90 \pm 0.06$ 
 II: $87.39 \pm 0.09$ | I: $97.73 \pm 0.03$ 
 II: $88.00 \pm 0.05$ | I: $97.70 \pm 0.04$ 
 II: $87.20 \pm 0.08$ | I: $97.45 \pm 0.05$ 
 II: $87.11 \pm 0.10$ | $84.69 \pm 0.32$ |
| 4 | I: $73.30 \pm 0.10$ 
 II $88.18 \pm 0.12$ | **I: $74.04 \pm 0.08$** 
 **II: $88.65 \pm 0.08$** | I: $74.21 \pm 0.10$ 
 II: $88.22 \pm 0.09$ | I: $73.39 \pm 0.12$ 
 II: $88.50 \pm 0.10$ | I: $73.44 \pm 0.10$ 
 II: $88.00 \pm 0.12$ | $86.12 \pm 0.14$ |
| 5 | I: $78.57 \pm 0.01$ 
 II $88.73 \pm 0.11$ | I: $78.66 \pm 0.03$ 
 II: $88.25 \pm 0.12$ | I: $79.07 \pm 0.02$ 
 II: $88.66 \pm 0.13$ | **I: $78.68 \pm 0.04$** 
 **II: $88.75 \pm 0.09$** | I: $78.01 \pm 0.04$ 
 II: $87.89 \pm 0.13$ | $86.46 \pm 0.17$ |

Table 2: Experimental results of ADE-Net with Kennedy Space Center dataset. Best performing results for each attack combination are in bold and the worst-performing is in red. The $\alpha_k$ hyperparameter is for the ensemble class cross-entropy, $\lambda_k$ is for the discriminator CKA and $\beta$ is for the discriminator attack cross-entropy.

| # Attack | All = 1.0 | $\lambda_k=\beta=0.1$ $\alpha_k=10.0$ | $\lambda_k=\beta=10.0$ $\alpha_k=0.1$ | $\lambda_k=\beta=1.0$ $\alpha_k$ Attack Dependent | $\lambda_k=0$ (No CKA) $\beta=\alpha_k=1.0$ | Baseline |
|---|---|---|---|---|---|---|
| 2 | I: $98.61 \pm 0.29$ 
 II: $95.84 \pm 0.59$ | **I: $98.87 \pm 0.22$** 
 **II: $96.24 \pm 0.54$** | I: $98.18 \pm 0.19$ 
 II: $95.28 \pm 0.61$ | I: $98.16 \pm 0.21$ 
 II: $95.38 \pm 0.49$ | I: $98.11 \pm 0.22$ 
 II: $95.22 \pm 0.53$ | $93.94 \pm 0.12$ |
| 3 | **I: $67.05 \pm 0.05$** 
 **II: $96.32 \pm 0.49$** | I: $66.81 \pm 0.06$ 
 II: $95.77 \pm 0.35$ | I: $67.24 \pm 0.05$ 
 II: $95.70 \pm 0.44$ | I: $67.20 \pm 0.05$ 
 II: $95.94 \pm 0.48$ | I: $67.03 \pm 0.07$ 
 II: $95.89 \pm 0.42$ | $95.15 \pm 0.05$ |
| 4 | I: $50.51 \pm 0.06$ 
 II: $96.02 \pm 0.16$ | I: $50.47 \pm 0.06$ 
 II: $97.21 \pm 0.09$ | I: $50.40 \pm 0.05$ 
 II: $96.10 \pm 0.11$ | **I: $50.52 \pm 0.04$** 
 **II: $97.24 \pm 0.08$** | I: $50.39 \pm 0.06$ 
 II: $95.89 \pm 0.14$ | $95.48 \pm 0.04$ |
| 5 | I: $55.66 \pm 0.03$ 
 II: $96.45 \pm 0.05$ | I: $55.70 \pm 0.04$ 
 II: $96.57 \pm 0.04$ | I: $55.45 \pm 0.03$ 
 II $96.35 \pm 0.05$ | I: $55.76 \pm 0.03$ 
 II: $96.42 \pm 0.04$ | **I: $54.97 \pm 0.03$** 
 **II: $96.63 \pm 0.04$** | $96.08 \pm 0.16$ |

Table 3: Experimental results of ADE-Net with the Houston dataset. Best performing results for each attack combination are in bold and the worst-performing is in red. The $\alpha_k$ hyperparameter is for the ensemble class cross-entropy, $\lambda_k$ is for the discriminator CKA and $\beta$ is for the discriminator attack cross-entropy.

