# OpenReview forum: "Improving Hyperspectral Adversarial Robustness Under Multiple Attacks"
_ICLR.cc/2023/TinyPapers — Submitted to Tiny Papers @ ICLR 2023_

### Official Review · Reviewer_8t8w · 2023-03-29

**Confidence:** 3

**Summary Of Contributions:**

The paper presents an Adversarial Discriminator Ensemble Network (ADE-Net) that addresses the vulnerability of semantic segmentation models classifying hyperspectral images to adversarial examples. ADE-Net employs a discriminator network to detect the attack type and a specialized ensemble network for each attack type.

**Rating:**

Clear, Correct, and Reproducible (CCR): a submission which meets the reviewing criteria

**Strengths And Weaknesses:**

Strong aspects:

1. The paper proposes an Adversarial Discriminator Ensemble Network (ADE-Net), which combines the strengths of adversarial training and adversarial detection to defend against adversarial attacks in one model.

2. The proposed model consists of two phases, with the Discriminator Phase adding adversarial detection via attack type classification and CKA to separate mixed attack data, and the Ensemble Phase constituted by a number of networks that focus on label classification.

3. The paper reports experimental results that demonstrate the effectiveness of ADE-Net in defending against various adversarial attacks on hyperspectral image classification.

**Suggested Changes:**

In Algorithm 1, there is a potential typo in the line "for b = bach1, . . . , bachB do", where "bach" should be replaced with "batch" to indicate that batches are being processed.

---

### Official Review · Reviewer_QdY9 · 2023-04-03

**Confidence:** 4

**Summary Of Contributions:**

The authors have proposed a method to improve adversarial robustness of semantic segmentation networks with hyperspectral images (HSI). The proposed adversarial discriminator ensemble network (ADE-Net) can achieve both attack type detection and network robustification under a unified model. More specifically, the discriminator network can classify the data by attack type and feed it into their specific attack-expert ensemble network.

**Rating:**

High Potential (HP): a submission which meets the reviewing criteria and has potential to make an impact on the field

**Strengths And Weaknesses:**

Strength:
1. Paper is clearly written. Motivation is also quite clear.
2. Comparison to other related methods seems adequate.
3. Introduction of an attack type discriminator is novel.

Weakness:
1. No comparison with other SOTA methods in the experiments.


**Suggested Changes:**

Add other SOTA comparisons in the experimental section.

---

### Author Response · Authors · 2023-05-30
**Opt-In for Archival**

All authors agree to Opt-In the paper "Improving Hyperspectral Adversarial Robustness Under Multiple Attacks" for Archival in ICLR 2023 TinyPapers

---

### Meta-Review · Area_Chair_HUon · 2023-04-06

**Recommendation:** Invite to archive
**Confidence:** 5

**Metareview:**

This paper investigates how to improve the robustness under multiple attacks and proposes an ADE-Net scheme. Generally, it is clear and interesting. However, the experiments lack of comparison with the SOTA results. Also, it would be helpful to test the proposed scheme's accuracy if the authors provide the source codes. Meanwhile, the idea is not novel.

**Summary:**

This paper improves the robustness of DNN under multiple attacks.

**Comments And Feedback To The Authors:**

Please refer to the meta review.

**Reason For Not Giving A Higher Recommendation:**

N/A

**Reason For Not Giving A Lower Recommendation:**

N/A

---

### Decision · Program_Chairs · 2023-04-08

Invite to archive